# Monoclonal Antibody against Porcine LAG3 Inhibits Porcine Reproductive and Respiratory Syndrome Virus Infection

**DOI:** 10.3390/vetsci11100483

**Published:** 2024-10-07

**Authors:** Hui Wang, Xu Zheng, Danyang Zheng, Xiaoqian Wang, Zhiqian Zhao, Mi Zhao, Qiang Guo, Yang Mu

**Affiliations:** 1Department of Preventive Veterinary Medicine, College of Veterinary Medicine, Northwest A&F University, Yangling 712100, China; w14789391337@163.com (H.W.); z15093333585@163.com (X.Z.); zzq597170371@163.com (Z.Z.); z18291105777@163.com (M.Z.); guoqiang@nwafu.edu.cn (Q.G.); 2Engineering Research Center of Efficient New Vaccines for Animals, Universities of Shaanxi Province and Ministry of Education, Yangling 712100, China

**Keywords:** inhibitory receptor, porcine LAG3, cell fusion, monoclonal antibody, PRRSV infection, co-culture system

## Abstract

**Simple Summary:**

Lymphocyte activation gene 3 (LAG3) is an inhibitory receptor, and the interaction between fibrinogen-like protein 1 and LAG3 is regarded as a new immune escape mechanism. Porcine reproductive and respiratory syndrome (PRRS) is an infectious disease seriously impacting the porcine industry worldwide. In this study, monoclonal antibodies (mAbs) against porcine LAG3 (*pLAG3*) were developed, and one mAb (1C2) showed good reactivity with *pLAG3* on PHA-activated porcine lymphocytes. The epitope recognized by mAb 1C2 was located at amino acid residues 214–435 of *pLAG3*. The LAG3 expression in the tissues of PRRSV-infected pigs increased remarkably. Interference of LAG3 expression on PHA-activated lymphocytes promoted PRRSV replication in the co-culture system of monocyte-derived dendritic cells and lymphocytes, whereas overexpression of LAG3 or blocking of the LAG3 signal with mAb 1C2 inhibited PRRSV replication, indicating that PRRSV infection activates the LAG3-signaling pathway, and this pathway plays an important role in PRRSV pathogenesis.

**Abstract:**

Lymphocyte activation gene 3 (LAG3) is an inhibitory receptor and the interaction between fibrinogen-like protein 1 and LAG3 can inhibit the anti-tumor effect of T cells both in vivo and in vitro, which was regarded as a new immune evasion mechanism. Porcine reproductive and respiratory syndrome (PRRS), caused by PRRSV, is an infectious disease characterized by reproductive disorders in pregnant sows and gilts and respiratory problems in pigs of all ages, seriously impacting the pig industry worldwide. In this study, monoclonal antibodies (mAbs) against porcine LAG3 (*pLAG3*) were developed, and one mAb (1C2) showed good reactivity with *pLAG3* on PHA-activated porcine peripheral blood lymphocytes. Epitope mapping showed the epitope recognized by mAb 1C2 was located at amino acid residues 214–435 of *pLAG3*. LAG3 expression in the tissues of PRRSV-infected pigs was detected, using mAb 1C2 as the primary antibody, and the results revealed that PRRSV infection caused a marked increase in LAG3 expression compared to the control group. Interference of LAG3 expression on PHA-activated lymphocytes promoted PRRSV replication in the co-culture system of monocyte-derived dendritic cells and lymphocytes, whereas overexpression of LAG3 or blocking of the LAG3 signal with mAb 1C2 inhibited PRRSV replication, indicating that PRRSV infection activates the LAG3-signaling pathway, suggesting that this pathway plays an important role in PRRSV pathogenesis. The results obtained lay the foundation for subsequent research on the role of LAG3 in PRRS and other diseases with persistent infection characteristics.

## 1. Introduction

Lymphocyte activation gene 3 (LAG3), also known as CD223, is a member of the immunoglobulin superfamily (IgSF) expressed on the surface of T cells, natural killer (NK) cells, NKT cells, plasmacytoid dendritic cells, and B cells [1,2]. As an immune checkpoint co-inhibitory molecule, LAG3 regulates T cell activation and proliferation, cytokine production, cytolytic activity, and immune cell homeostasis by delivering inhibitory signals [1,3]. Major histocompatibility complex (MHC) class II, liver sinusoidal endothelial cell lectin, galactose lectin 3, and α-synuclein are the ligands identified for LAG3 [4]. Fibrinogen-like protein 1 (FGL1) has been reported to be the main inhibitory ligand of LAG3 in the FGL1–LAG3 pathway, which is considered a promising immunotherapeutic target [5]. In a knockout mouse model, LAG3 impeded T cell expansion and controlled the number of memory T cells [6]. When T cells are activated to a certain extent, immunosuppressive molecules, such as LAG3, programmed death 1 (PD-1), and cytotoxic T lymphocyte antigen 4 (CTLA-4), are expressed to maintain the immune response in a stable state. When the cell homeostasis regulated by LAG3- is perturbed, immune-mediated tissue damage occurs rapidly. Furthermore, persistent antigen stimulation, such as in chronic viral infections and cancer, increases chronic LAG3 expression, leading to T cell exhaustion and the subsequent impairment of T cell function [7]. Guy et al. reported that LAG3 moves to the immunological synapse and is associated with the T cell receptor (TCR)–CD3 complex in CD4^+^ and CD8^+^ T cells in the absence of binding to MHC II [8]. LAG3 is also considered a marker of regulatory T cell (Treg) subpopulation activation [9]. On activated T cells, LAG3 reduces cytokine and granzyme production and proliferation while promoting Treg differentiation [10].

LAG3 is the third inhibitory receptor to be targeted in the clinic and is the most promising immune checkpoint after CTLA-4 and PD-1 [11]. It controls excessive activation following persistent antigen exposure to prevent the development of autoimmunity; however, it can also contribute to T cell dysfunction in the tumor microenvironment (TME) [12]. Porcine reproductive and respiratory syndrome (PRRS), caused by the PRRS virus (PRRSV), is one of the infectious diseases characterized by reproductive disorders in pregnant sows and gilts and respiratory problems in pigs of all ages, causing the most serious impact on the pig industry worldwide. The PRRSV genome, about 15 kb long, includes at least 11 open reading frames (ORFs). Based on ORF5 classification, PRRSV is divided into two species: *Betaarterivirus suid* 1 or PRRSV-1 and *Betaarterivirus suid* 2 or PRRSV-2 [13]. Ruedas-Torres et al. evaluated the expression of immune checkpoints in the thymus, lungs, and tracheobronchial lymph nodes (TBLN) of piglets infected with two PRRSV-1 strains of different virulence and found a similar progressive increase in LAG3 gene expression in the lungs and TBLN of both infected groups. However, a remarkable upregulation was observed in the thymi of both infected groups at 6 days post-infection (dpi), and the level remained high, especially in those infected with the virulent Lena strain [14]. Moreover, this upregulation was associated with disease progression, a high viral load and cell death [15].

In 1990, Triebel et al. isolated nine genomic clones from a human genomic DNA library, using an FD19 probe, and found that the length of the LAG3 gene was approximately 6.6 kb and contained eight exons encoding a type I transmembrane protein consisting of 498 amino acids [16]. In humans, the gene-encoding LAG3 shares a common intron and exon with the gene encoding CD4, and the amino acid sequence is partially identical (>20%) [17]. LAG3 is composed of three parts: extracellular, transmembrane, and cytoplasmic regions. Its extracellular region consists of four domains called D1–D4, with D4 being closest to the cell surface. The KIEELE motif in the cytoplasmic region is an important site for MHCII–LAG3 signaling, and the deletion of this region prevents it from performing its inhibitory biological function [18]. Mature LAG3 in mice is cleaved by metalloproteinases and broken into a 54 kDa soluble p54 extracellular fraction and a 16 kDa p16 transmembrane-cytoplasmic fraction [19]. Several regions in the LAG3 protein structure are involved in intracellular storage, including the D1 and D2 regions, which can affect the intracellular storage of LAG3 molecules [20]. Kim et al. amplified the gene encoding porcine LAG3 (*pLAG3*) from phytohemagglutinin (PHA)-stimulated porcine spleen cells, which is 1524 bp long and encodes 507 amino acids [21]. Mice antisera against recombinant *pLAG3* containing D1 and D2 structural domains expressed in CHO cells, and when PHA-stimulated porcine splenocytes were examined by Western blotting, the molecular mass of LAG3 detected was about 67 kDa.

Viral invasion induces immune dysregulation, in which virions escape clearance by the host immune system and promote replication by interacting with the host [22]. FGL1 is the main inhibitory ligand of LAG3, and our previous study indicated that PRRSV infection resulted in a marked increase in FGL1 in the sera of pigs [23]; therefore, we speculated that LAG3 might play an important role in the pathogenesis of PRRSV, as previously suggested [15]. To date, the study of LAG3 has mainly focused on humans and mice. There are few reports on the function of LAG3 in pigs, and its role in the process of PRRSV infection has been rarely studied. Owing to the lack of corresponding antibodies, there have been no studies on the correlation between LAG3 and PRRSV replication.

In this study, we cloned and expressed *pLAG3* and prepared the corresponding mAbs to clarify the role of LAG3 in the pathogenesis of immunosuppressive infectious diseases in pigs.

## 2. Materials and Methods

### 2.1. Virus Strains, Cells, Vectors, Antibodies, Tissues, and Animals

The highly pathogenic PRRSV-2 strain SD16 was isolated and sequenced in our laboratory (GenBank: ID JX087437.1). The highly pathogenic PRRSV-2 strain XJA1 (HP-PRRSV-XJA1) (GenBank: ID FEF112445.1), SP2/0 and HEK-293T cell lines, pET-28a, pET-21b, pcDNA3.1/V5-HisB, pCMV-Flag vectors, mAb2-5G2 (an anti-idiotypic mAb with IgG1 heavy chain and κ light chain) [24], mAb against swine leukocyte antigen (SLA)-DRα [25,26], and porcine HEV 239 protein were preserved in our laboratory. Tissue samples of the lungs, liver, spleen, and mesenteric lymph nodes (MLNs) from mock and PRRSV-infected pigs were prepared and stored in our laboratory. Six- to eight-weeks-old female BALB/c mice were purchased from Chengdu Dossy Experimental Animals Co., Ltd., and PRRSV-free pigs were purchased from the Xionglong Pig Farm in Yangling Agricultural High-tech Industry Demonstration Zone, Shaanxi Province, Chengdu, China. The animal protocol was reviewed and approved by the Animal Welfare Committee of Northwest A&F University.

### 2.2. Isolation of Porcine Peripheral Blood Mononuclear Cells (PBMCs)

PBMCs were purified from fresh anticoagulated blood of PRRSV-free pigs according to the instructions of lymphocyte separation medium (Cat# P8770, Solarbio, Beijing, China), washed two times with phosphate-buffered saline (PBS), then suspended in a cell-freezing medium, and stored in liquid nitrogen or RPMI 1640 medium (Cat# 31800022, ThermoFisher, Waltham, MA, USA) containing 10% fetal bovine serum (FBS, Cat# S711-001, Lonsera, Ciudad de la Costa, Uruguay), 100 μg/mL streptomycin, and 100 U/mL penicillin (Cat# 15140122, ThermoFisher). The concentration was adjusted to 1 × 10^6^ cells/mL and cultured in 6-well plates (2 × 10^6^ cells/well) in the presence of PHA (2 μg/mL, Cat# L1668, Sigma-Aldrich, St. Louis, MO, USA) for 3 days at 37 °C in 5% CO_2_. 

### 2.3. Expression and Purification of pLAG3

A pLAG3 recombinant protein (37–435 aa) was expressed and purified. Briefly, the total RNA of the collected PBMCs was extracted using RNAiso Plus reagent (Cat# 9108, Takara Bio (Beijng), Beijing, China), then cDNA was synthesized using a PrimeScript^TM^ RT reagent kit (Cat# RR037A, Takara Bio). The *pLAG3* gene (encoding 37–435 aa) was amplified with primers *pLAG3*-109-peF and *pLAG3*-1305-peR (Table 1) using PrimeSTAR^®^ Max DNA polymerase (Cat# R045A, Takara Bio) according to the manual’s instructions. The recovered PCR product was ligated to pET-28a vector and recombinant plasmid pET28a-*pLAG3* was transformed into *E. coli* BL21(DE3)-competent cells (Cat# CD601-02, TransGen, Beijing, China). The recombinant pLAG3 protein was obtained by adding 1 mmol/L IPTG (Cat# 9030, Takara Bio) and inducing 6 h at 37 °C when the A_600nm_ of cultures reached 0.6–0.8. The *E. coli* cell pellet was harvested by centrifugation for 10 min at 4 °C and 10,000× *g*. The pellet obtained from 1 L of culture and resuspended in 50 mL PBS was sonicated on ice. After centrifugation for 10 min at 4 °C and 3000× *g*, the precipitate was resuspended in solubilization buffer B (100 mmol/L Na_2_PO_4_, 10 mmol/L Tris-HCl, 8 mol/L urea, pH 8.0) and the recombinant pLAG3 protein was purified with cOmplete™ His-Tag Purification Resin (Cat# 5893682001, Roche, Basel, Switzerland). Expression and purification effects were analyzed by SDS-PAGE and Western blotting. Finally, the recombinant pLAG3 protein was dialyzed, and the dialyzed sample was concentrated using an Amicon Ultra centrifugal Filter Unit (Cat# UFC9010, Millipore, Billerica, MA, USA) with a molecular weight cut-off of 10 kDa. The concentration of the recombinant pLAG3 protein was determined using a Pierce^TM^ BCA Protein Assay Kit (Cat# 23227, ThermoFisher) and stored at −20 °C for next use.

### 2.4. Production of mAbs against pLAG3

To produce hybridoma cells secreting *pLAG3* specific mAbs, purified recombinant pLAG3 protein (50 μg/mouse) was emulsified with Freund’s complete adjuvant (Cat# F5881, Sigma-Aldrich) in primary immunization by subcutaneous injection. Freund’s incomplete adjuvant (Cat# F5506, Sigma-Aldrich) was used in the three subsequent immunizations at 2-week intervals, using the same method. Serum antibody titers of mice were detected using an indirect enzyme-linked immunosorbent assay (iELISA) on the tenth day after the fourth immunization. For iELISA detection, 96-well plates (BS-PS-96W, Biosharp, Labgic Tech, Anhui, China) were coated with 0.4 μg recombinant *pLAG3* per well overnight at 4 °C. After washing three times (5 min/time) with PBS containing 0.05% Tween-20 (PSB’T, *v*/*v*) and blocking with PBS’T containing 5% skim milk (Cat# 1172GR500, BioFroxx, Munich, Germany) (SM-PSB′T, *w*/*v*), diluted serum samples (100 μL/well) were added and incubated for 60 min at 37 °C. The secondary antibody was a peroxidase-conjugated AffiniPure Goat Anti-Mouse IgG (H + L) (Cat# 115-035-003, Jackson ImmunoResearch, Lancaster, PA, USA) diluted 1:5000, and the condition was 60 min at 37 °C. The substrate was 3,3′,5,5′-tetramethylbenzidine (Cat# 860336, Sigma-Aldrich) with an incubation time of 15 min at room temperature (RT) in the dark. A Microplate Absorbance Reader (Bio-Rad, Hercules, CA, USA) was used to read optical density values at 450 nm (OD450).

The mouse with the highest serum antibody titer received a booster immunization via intraperitoneal injection on the eleventh day after the fourth immunization. Three days after the boost, the mouse was euthanized, and the anti-*pLAG3* serum was separated. SP2/0 cells were fused to isolated mouse splenocytes. The RPMI 1640 medium containing HAT/HT (Cat# H0262/H0137, Sigma-Aldrich) and 10% FBS was used for screening. The screening system was the same as the iELISA method described above, except that the primary antibody used was the cell culture supernatant. After the positive clones were systematically identified, ascites were prepared, and antibody purification and concentration determination were performed [23].

### 2.5. pcDNA3.1/V5-HisB-flpLAG3 Recombinant Plasmid Construction 

Using the primers pLAG3-eeF and pLAG3-eeR (Table 1), the full-length gene of *pLAG3* (flpLAG3) was amplified. The recovered PCR product was ligated to the pcDNA3.1/V5-HisB vector, and the correct recombinant plasmid was named pcDNA3.1/V5-HisB-flpLAG3.

### 2.6. Indirect Immunofluorescence Assay (IFA)

The pcDNA3.1/V5-HisB-flpLAG3 plasmid-transfected HEK-293T cells were fixed in 4% paraformaldehyde in PBS for 10 min at RT. The cells were washed three times with PBS, permeabilized with 0.25% Triton X-100 in PBS for 15 min, washed again, and blocked with 1% BSA/PBS. After washing, cells were incubated with in-house anti-porcine LAG3 primary antibodies or anti-His mAb for 1 h at 37 °C, then incubated with Cy3-conjugated AffiniPure Goat Anti-Mouse IgG (H + L) (Cat# 115-165-003, Jackson ImmunoResearch) for 1 h at 37 °C. After being stained with DAPI, the cells were visualized with a Leica microscope system (Leica AF6000, Munich, Germany).

### 2.7. Western Blotting Detection

At 48 h post-transfection (hpt), pcDNA3.1/V5-HisB-flpLAG3 plasmid-transfected HEK-293T cells were washed twice with PBS and harvested in NP40 lysis buffer (Cat# P0013F, Beyotime, Shanghai, China) containing 1 mmol/L PMSF (Cat# ST506, Beyotime). A Pierce^TM^ BCA Protein Assay Kit (Cat# 23227; ThermoFisher) was used to determine protein concentrations in the samples. The cell lysates were resolved by SDS-PAGE, and a Bio-Rad Mini Trans-Blot electrophoretic transfer unit was used to transfer the samples onto PVDF membranes (Cat# ISEQ00010, Millipore). The membrane was blocked with SM-PSB′T at RT and incubated with in-house anti-pLAG3 primary antibodies or ProteinFind^®^ anti-His mouse mAb (Cat# HT 501, TransGen) or ProteinFind^®^ anit-Flag mouse mAb (Cat# HT201, TransGen) overnight at 4 °C. After washing, horseradish peroxidase (HRP)-linked anti-mouse IgG (Cat# 7076, CST, Danvers, MA, USA) or HRP-linked anti-rabbit IgG (Cat# 7074, CST) was added and incubated for 1 h at RT. Naïve mice sera were used as a negative control. Beta-tubulin (β-tubulin) (ProteinFind^®^ anti-β-tubulin mouse mAb, Cat# 101-01, TransGen) or GAPDH (GAPDH Rabbit mAb, Cat# 2118, CST) was used as the loading control. Imprinted bands were visualized using a ChemiDoc^TM^ MP Imaging System (Bio-Rad).

### 2.8. ELISA Additivity Test

An ELISA additivity test was used to identify the epitopes recognized by the mAbs [23,27,28]. An additive index (AI) of mAbs was calculated using the following formula:AI = ((2 × A_1 + 2_/(A_1_ + A_2_)) − 1) × 100%
where A_1_ refers to the assayed OD of the first mAb, A_2_ refers to the assayed OD of the second mAb, and A_1 + 2_ refers to the assayed OD when adding equal amounts of the two mAbs to the same well. If the same epitope was bound by both mAbs, the AI would be negligible; otherwise, the AI would be near 100 if the two mAbs bound to topographically unrelated the two epitopes.

### 2.9. Identification of the Antigen Region Recognized by mAb 1C2

Based on the *pLAG3* gene sequence we obtained (GenBank ID: MK813967.1), the possible B cell epitopes were predicted using the ABCpred website (www.imtech.res.in/raghava/abcpred/, accessed on 6 October 2021). Combined with the epitope distribution results predicted by the Protean module of LaserGene software (DNASTAR, Inc., v 17.6), *pLAG3* (37–435 aa) was divided into three overlapping fragments for expression: pLAG3-1 (37–175 aa), pLAG3-2 (168–326 aa), and pLAG3-3 (215−435 aa).

The primers pairs pLAG3-109-peF and pLAG3-525-peR, pLAG3-502-peF and pLAG3-978-peR, and pLAG3-643-peF and pLAG3-1305-peR were used to amplify the corresponding gene fragments, which were then inserted into the pET-21b vector. The pLAG3-3 (215−435 aa) gene was simultaneously amplified using the primers pLAG3-643-eeF and pLAG3-1305-eeR and inserted into the pCMV-Flag vector for eukaryotic expression. The reactivity of the truncated protein with mAb 1C2 was determined by Western blotting.

### 2.10. Detection of LAG3, SLA-DRα, and FGL1 Expression in Tissues of PRRSV-Infected Pigs

Lung, liver, spleen, and MLN tissue samples were collected from mock and PRRSV-infected pigs. All tissue samples were derived from at least three pigs. The sexes of these pigs were random, and they were tested negative for PRRSV, CSFV, PCV2, ASFV, and antibodies against PRRSV and ASFV at 4 weeks old. The 5 pigs in the PRRSV-infected group were inoculated with 0.5 mL HP-PRRSV-XJA1 with 1 × 10^5^ 50% tissue culture infectious dose/mL (TCID_50_/mL) via intranasal routes and 0.5 mL via neck intramuscular injection at 9 weeks old. The 5 pigs in the mock group were inoculated with 1 mL DMEM medium using the same method. Tissue samples were collected 21 days post-infection. The mock piglets appeared normal, whereas the PRRSV-infected piglets demonstrated extensive pneumonia and viremia when they were euthanized and sampled [29]. RNA was extracted from these tissues to detect viral load and target gene transcription. NP40 lysis buffer (Cat# P0013F, Beyotime) containing 1 mmol/L PMSF (Cat# ST506, Beyotime) was added to these tissues for grinding, and then the supernatant was obtained after centrifugation for 20 min at 4 °C and 3000× *g*. Western blotting analysis was performed for the prepared samples to determine the effect of PRRSV infection on the expression of target proteins in pig tissues.

### 2.11. Quantitative Real-Time PCR (qPCR)

qPCR was performed with 2×RealStar Green Power Mixture with ROX (Cat# A313, GenStar, Shenzhen, China) and a Step One Plus^TM^ Real-Time PCR system (Applied Biosystems, Carlsbad, CA, USA). The reaction procedure is 10 min at 95 °C, then 40 cycles of 15 s at 95 °C and 1 min at 60 °C. GAPDH was used as an internal control. The primer sequences are listed in Table 1. Relative quantification of *pLAG3* gene expression was calculated with the 2^−ΔΔCt^ method. To quantify the PRRSV load, nine serial 10-fold dilutions of a recombinant plasmid containing the 144 bp PRRSV ORF7 gene preserved in our laboratory [30] were introduced into each plate to test inter-run variations. To avoid overestimating the number of viral particles, the results of the PRRSV load were shown as quantification cycle (Cq) values [14,15,31]. All samples were analyzed in triplicate.

### 2.12. Induction of Monocyte-Derived Dendritic Cells (MoDCs)

Porcine PBMCs (5 × 10^6^ porcine PBMCs in 1 mL RPMI-1640 medium with 10% FBS) were plated in a 12-well plate for 4 h, then the non-adherent cells were removed. The adhered cells, namely monocytes, were cultured in 1 mL RPMI-1640 containing 10% FBS, 20 ng/mL recombinant porcine IL-4 (rp IL-4, R&D Systems, Minneapolis, MN, USA) and 20 ng/mL recombinant porcine granulocyte–macrophage colony-stimulating factor (rp GM-CSF, R&D Systems), referred to as DCs medium, for 7 days. Every two days, the culture medium was replaced with fresh DC medium, and cell morphological changes were observed under a light microscope [32]. The induced MoDCs were harvested, and the expression of the SLA-DRα chain was analyzed by Western blotting or infected by 0.1 MOI PRRSV SD16 and harvested at 48 h post-infection (hpi) to analyze PRRSV load and N protein expression.

### 2.13. Establishment of a Co-Culture System

A co-culture system consisting of porcine lymphocytes and MoDCs was established. Briefly, porcine PBMCs were induced to differentiate into MoDCs as described above, and the MoDCs were then infected with the PRRSV SD16 strain at 1 MOI. On the other hand, porcine lymphocytes were plated in 12-well plates at 3 × 10^6^ cells per well, then PHA was added at a final concentration of 4 μg/mL to activate the lymphocytes. After 24 h of stimulation, the pcDNA3.1/V5-HisB-flpLAG3 recombinant plasmid or pcDNA3.1/V5-HisB plasmid was transfected using lipofectamine^TM^ 3000 (Cat# L3000008, ThermoFisher). The transfected cells were collected at 48 h after transfection and co-cultured with PRRSV-infected MoDCs at a 1:1 ratio; the PHA-simulated lymphocytes were transfected with LAG3-siRNA684 (Table 2, Genepharma, Shanghai, China) and were collected and co-cultured with PRRSV-infected MoDCs at a 1:1 ratio at 36 h after transfection; or the simulated lymphocytes were incubated with 50 μg/mL mAb 1C2 at 37 °C for 1 h, and then were co-cultured with PRRSV-infected MoDCs at a 1:1 ratio. The cells and supernatants were collected after 24 and 48 h of co-culture. The cells were used to detect the copy number of the PRRSV N gene by reverse transcription (RT)-qPCR and the expression level of N protein by Western blotting, and the supernatant was used to detect the virus titer.

### 2.14. Statistical Analysis

All data were verified in three independent experiments and were presented as mean ± standard deviation (SD). A statistical analysis was performed to examine the differences in PRRSV load and relative mRNA expression levels of *pLAG3*. The data were analyzed using GraphPad Prism software 8.4 (GraphPad Software, San Diego, CA, USA) with an ordinary one-way ANOVA, followed by a Tukey’s test to compare the mean of each column with the mean of every other column. Comparisons between groups were considered statistically significant at *p* < 0.05.

## 3. Results

### 3.1. pLAG3 Gene Amplification and pLAG3 Protein Expression 

The *pLAG3* gene, without the fragment encoding the signal peptide, was amplified, and the length of the target gene was 1200 bp (Figure 1A). The sequence was uploaded to NCBI, and GenBank ID ‘MK813967.1’ was obtained. The recombinant plasmid pET28a-*pLAG3* was correctly identified and transformed into *E. coli* BL21(DE3)-competent cells. The recombinant pLAG3 protein was expressed mainly as inclusion bodies and was approximately 45 kDa (Figure 1B). Using ProteinFind^®^ anti-His mouse mAb as the primary antibody, the expression of recombinant *pLAG3* with His tag was identified by Western blotting (Figure 1C). Relatively purified recombinant pLAG3 protein was obtained after purification with cOmplete™, Complete Applications, Inc. San Francisco, CA, USA. His-Tag Purification Resins (Figure 1D).

### 3.2. Generation and Characterization of pLAG3 mAbs

After the fourth immunization, serum antibody titers of the immunized BALB/c mice were measured by iELISA. The results showed that one mouse had an antibody titer of 1:128,000 (Figure 2A). The spleen of this mouse was harvested to prepare a splenocyte suspension. After fusion and screening, two hybridoma cell lines, 1C2 and 3E11, which stably secreted antibodies against *pLAG3*, were obtained. The reactivity of mAbs 1C2 and 3E11 with *pLAG3* expressed by pcDNA3.1/V5-HisB-flpLAG3-transfected HEK-293T cells was identified by Western blotting and IFA. As shown in Figure 2B (The original images were provided in Appendix A) and Figure 2C, both mAbs specifically reacted with *pLAG3* expressed by the transfected HEK-293T cells. Hybridoma cells of 1C2 and 3E11 were injected intraperitoneally into liquid paraffin-pretreated mice, and ascites were collected. The antibody titers of the ascites reached 1:1,048,576 and 1:262,144, respectively (Figure 2D). Both heavy and light chains of mAbs 1C2 and 3E11 are IgG1 and κ chains, as indicated by the detection results of the IsoStrip^TM^ Mouse Monoclonal Antibody Isotyping Kit (Cat# 11493027001, Roche) (Figure 2E) (The original images were provided in Appendix A). The saturation curves of the antigen with each antibody were shown in Figure 2F. The detection values for both mAbs plateaued at 1:1024 dilution. Then, using a 1:1000 dilution, an ELISA additivity test was performed, and the AI was 3.50% (Table 3), indicating that the two mAbs had no additive effect, and that the mAbs may recognize the same epitope. mAb 1C2 was coarsely extracted with saturated ammonium sulfate solution and then further purified by Protein G (Cat# L00209, Genscript, Nanjing, China) affinity chromatography. SDS-PAGE analysis showed that purified mAb 1C2 was obtained (Figure 2G) (The original image was provided in Appendix A).

### 3.3. The Antigen Region Recognized by mAb 1C2 Located at 214–435 AA of pLAG3

To identify the antigen region recognized by mAb 1C2, *pLAG3* (37–435 aa) was divided into three overlapping fragments for expression based on the possible B cell epitopes predicted using the online software ABCpred, version 10, and the epitope distribution predicted using LaserGene software (Figure 3A). The gene fragments encoding these peptides were amplified and inserted into the pET-21b vector for prokaryotic expression. The peptide pLAG3-1 was not expressed. The tagged control protein (porcine HEV 239 protein) and the expressed pLAG3-2 and pLAG3-3 proteins were detected by Western blotting using anti-His mouse mAb as the primary antibody. However, mAb 1C2 only reacted with pLAG3-3 (Figure 3B) (The original images were provided in Appendix A). The gene fragment encoding pLAG3-3 was inserted into the pCMV-Flag eukaryotic expression vector to construct the recombinant plasmid, pcMV-Flag-pLAG3-3, which was then transfected into HEK-293T cells. Using purified mAb 1C2 as the primary antibody, ProteinFind^®^ anti-Flag mouse mAb as the positive control antibody, and cells transfected with the pCMV-FLAG vector as the negative controls, the Western blotting results showed that mAb 1C2 specifically reacted with the pLAG3-3 protein expressed by HEK-293T cells at 48 hpt (Figure 3C) (The original images were provided in Appendix A).

### 3.4. PRRSV Infection Promotes LAG3, SLA-DRα, and FGL1 Expression In Vivo 

To determine the effect of PRRSV infection on the FGL1/SLAII-LAG3 pathway in vivo, the total RNA and protein of the lung, liver, spleen, and MLN samples were prepared after grinding the samples at low temperatures. The lung, the target organ of PRRVS, and the spleen and lymph nodes, both peripheral immune organs, had a higher viral load, and there was also a certain amount of virus in the liver tissue, although it was not as obvious as in the above three tissues (Figure 4A). The expression of *pLAG3* mRNA and protein was analyzed by RT-qPCR and Western blotting, using homemade mAb 1C2 as the primary antibody. Its level in the tissues of PRRSV-infected pigs increased, especially in the lungs and spleen (Figure 4B,C). The expression of SLA-DRα mRNA and protein in these four tissues was analyzed by RT-qPCR and Western blotting, using a homemade mAb2E11D9 as the primary antibody. Unlike the mRNA of both SLA-DRα and SLA-DRβ, which were significantly inhibited by PRRSV replication, the SLA-DRα protein expression increased after PRRSV infection in DC [26], both mRNA and protein expression of SLA-DRα were increased in these tissues after PRRSV infection (Figure 4D,E). The FGL1 mRNA transcript levels and protein levels were analyzed by RT-qPCR and Western blotting, using a homemade mAb 4G7 as the primary antibody in PRRSV-infected liver tissue and found that the levels increased (Figure 4F,G).

### 3.5. Monocytes Were Induced into MoDCs 

Porcine monocytes were isolated from porcine PBMCs by a differential adherence method. When rp GM-CSF and rp IL-4 were added to the culture medium, small protrusions appeared on the cell surface after 3 days. On the fifth day, a large number of dendritic protrusions of different lengths and thicknesses appeared on the cell surface, and the cells showed typical morphological characteristics of dendritic cells. On day 7, dendritic protrusions on the cell surface were more prominent (Figure 5A). The expression of SLA-DRα was markedly increased when analyzed by Western blotting (Figure 5B). When infected, induced MoDCs with PRRSV, PRRSV viral load, and N protein expression increased significantly compared with that of monocytes (Figure 5C,D). Since MHC II molecules were mainly expressed in antigen-presenting cells, it was suggested that MoDCs were obtained by differential adherent purification and induction culture with rp GM-CSF and rp IL-4.

### 3.6. PRRSV Replication Increases in the Co-Culture System When Lymphocytes Overexpress LAG3 

To determine the effect of LAG3 overexpression by lymphocytes on PRRSV replication, porcine peripheral blood lymphocytes were stimulated with PHA for 24 h and then transfected with pcDNA3.1/V5-HisB-flpLAG3 plasmid. The cells were collected at 24 and 48 hpt. Western-blotting analysis was performed using mAb 1C2 as the primary antibody, and the results showed LAG3 expression increased significantly at 48 hpt (Figure 6A). MoDCs infected with 1 MOI PRRSV SD16 were co-cultured with pcDNA3.1/V5-HisB-flpLAG3-transfected lymphocytes for 48 h at a ratio of 1:1. At 24 and 48 hpi, co-cultured cells and supernatants were collected to detect LAG3 expression and PRRSV replication. Compared to those in the control group transfected with the pcDNA3.1/V5-HisB plasmid, LAG3 protein expression increased markedly at both 24 and 48 hpi (Figure 6B). The PRRSV N gene copy number in the cells increased significantly at 24 and 48 hpi (Figure 6C), and the N protein expression also increased dramatically (Figure 6B). Although the viral titer in the supernatant was not significantly different from that in the control group at 24 hpi, it was significantly higher than that in the control group at 48 hpi (Figure 6D). These results indicate that PRRSV replication increased in the co-culture system when LAG3 was overexpressed by lymphocytes.

### 3.7. PRRSV Replication in the Co-Culture System Decreases with Interference of Lymphocyte LAG3 Expression by siRNA

To determine the effect of interference of lymphocyte LAG3 expression on PRRSV replication, siRNA684-, siRNA987-, and siRNA1347-targeting *pLAG3* and a negative control siRNA were synthesized, and porcine lymphocytes activated by PHA for 24 h were transfected. Samples collected at 24 hpt were analyzed. The results showed that the interference effect of siRNA684 on LAG3 expression was the most significant (Figure 7A,B). Next, 1 MOI PRRSV SD16-infected MoDCs was co-cultured with lymphocytes transfected with siRNA684 for 24 h at a ratio of 1:1. Cell and supernatant samples were collected at 24 and 48 hpi, and LAG3 expression and PRRSV replication were analyzed. Compared to the negative control siRNA transfection group, siRNA684 transfection significantly interfered with LAG3 mRNA expression in lymphocytes in the co-culture system (Figure 7C). The expression of LAG3 protein (Figure 7D), the PRRSV N gene copy number (Figure 7E) and the N protein expression (Figure 7D) decreased markedly at 24 and 48 hpi. The results showed that the viral titer in the culture supernatant of the experimental group transfected with siRNA684 also decreased significantly at 24 and 48 hpi (Figure 7F). These results indicate that PRRSV replication was significantly inhibited when LAG3 expression in lymphocytes was inhibited in the co-culture system.

### 3.8. PRRSV Replication in the Co-Culture System Decreases When the LAG3 Signal Is Blocked by Specific mAb 

The mAb 1C2 prepared in this study is specifically bound to LAG3 on the surface of porcine lymphocytes. We speculated that pretreatment of PHA-activated lymphocytes with this mAb could block the inhibition signal transmitted by LAG3 to activated lymphocytes, thus maintaining the killing function of activated lymphocytes and inhibiting PRRSV replication in the co-culture system. To confirm this hypothesis, we added mAb 1C2 at a final concentration of 0.05 mg/mL to the culture medium of porcine lymphocytes stimulated with PHA for 24 h. After incubation at 37 °C for 1 h, the collected cells were co-cultured with 1 MOI PRRSV-infected MoDCs in a ratio of 1:1. Cell and supernatant samples were collected at 24 and 48 hpi. The results showed that compared to the group of isotype antibody control, the PRRSV N gene copy number in the group of mAb 1C2 treatment group decreased significantly at 24 and 48 hpi (Figure 8A), and N protein expression also decreased significantly (Figure 8B). The viral titer in the culture supernatant was slightly decreased at 24 hpi, but the difference was not significant compared to that of the control group, which was decreased markedly at 48 hpi (*p* < 0.05) (Figure 8C). These results indicated that PRRSV replication was inhibited when LAG3-specific mAbs were added to the co-culture system to block the transmission of the LAG3-inhibitory signal.

## 4. Discussion

LAG3 (CD223) inhibits the activation of host immune cells and promotes suppressive immune responses [3]. Normally, LAG3 expressed on NK cells and activated T cells helps maintain immune homeostasis and prevents the occurrence of immune overreaction or autoimmune diseases. However, in TME or chronic infectious diseases, the continuous activation of T cells results in the co-expression of co-inhibitory receptors, which leads to incapacitation and even apoptosis of T cells. This is a “deceptive mechanism” by which tumor cells and pathogens of chronic infectious diseases can escape from being killed by the host immune system [33]. Currently, LAG3 in humans and mice has been well studied, and human LAG3 is now regarded as the most promising immune checkpoint after PD-1 and CTLA-4 [34]. However, *pLAG3* has not been studied extensively. In this study, we developed and validated two specific mouse mAbs against recombinant *pLAG3*, mAb 1C2, which recognizes an antigenic region at amino acid residues 214–435 of *pLAG3*. Using in-house mAb 1C2 and the established co-culture system, we analyzed the effect of PRRSV, an immunosuppressive pathogen of pig disease, on the expression of *pLAG3* in vivo and the effects of regulating LAG3 expression and blocking LAG3 signaling on PRRSV replication in vitro. One of the main features of PRRS is the persistent viral infection in lymphoid tissues. We found that PRRSV infection resulted in a remarkable increase in *pLAG3* expression in tissue samples from the lungs, liver, spleen, and MLNs, which, combined with our previous findings of elevated FGL1 levels in serum samples of PRRSV-infected pigs, suggested that PRRSV infection activated the FGL1–LAG3 pathway.

LAG3 is composed of extracellular, transmembrane, and intracellular regions [35]. To obtain the *pLAG3* recombinant protein expressed in *the E. coli* system, we cloned the coding sequence of the *pLAG3* extracellular region (encoding 37–435 aa) without the sequence encoding of the signal peptide. The Western blotting analysis of the pLAG3 recombinant protein revealed faint imprinted bands in the lane of the uninduced sample and the supernatant of cell lysate (Figure 1C), indicating that the pLAG3 recombinant protein has background expression in the uninduced sample and partly expressed in soluble form. None of these factors affected the acquisition of purified recombinant *pLAG3* (Figure 1D). The amount of target protein expressed in the soluble form was too low (Figure 1B); therefore, we used the target protein expressed in the inclusion form. Although mice were immunized with renatured inclusion proteins, specific fluorescence was observed in PHA-stimulated porcine PBMCs when sera from immunized mice were used as primary antibodies in the IFA analysis. Therefore, we fused SP2/0 cells with splenocytes from immunized mice and prepared mAbs against *pLAG3* using hybridoma technology. Finally, two mAbs, 1C2 and 3E11 (Figure 2), were obtained. 

The study of antigenic epitopes is important for the diagnosis and treatment of infectious diseases and vaccine development, and it has significant value in research on immune-system-related diseases [36]. Combined with mAb 1C2 presenting a better binding capability than mAb 3E11 and the result of the ELISA additivity test (Figure 2D, F and Table 3), *pLAG3* (37–435 aa) was divided into three overlapping fragments for expression to identify the antigen region recognized by mAb 1C2, according to the possible B cell epitopes predicted using online software ABCpred and the epitope distribution predicted using LaserGen software, v 17.6. Several prokaryotic expression vectors were used in the experiment; however, pLAG3-1 was not successfully expressed. mAb 1C2 reacted with both prokaryotic- and eukaryotic-expressed pLAG3-3 but not with pLAG3-2 or the control protein (Figure 3B, C). Therefore, the epitope recognized by mAb 1C2 is located in amino acid residues 214–435 of *pLAG3*. Subsequently, it was necessary to determine the exact position of the epitope recognized by mAbs 1C2 and 3E11 by truncating and synthesizing short peptides.

LAG3 mRNA level in the lung, liver, spleen, and MLN tissues of PRRSV-infected pigs increased when analyzed using RT-qPCR (Figure 4B), and LAG3 protein expression in these tissues was upregulated when mAb 1C2 was used as the first antibody in the Western-blotting analysis, especially in the lung and spleen tissues (Figure 4C). Ruedas-Torres et al. infected 5-week-old Landrace×Large White piglets with two PRRSV-1 strains (PRRSV Lena strain with virulence and PRRSV-13249 strain with low virulence) and analyzed LAG3 gene expression in lung tissues and TBLN by RT-qPCR at 1, 3, 6, 8, and 13 dpi. LAG3 expression showed a similar progressive increase in both infected groups, especially in the lungs of piglets infected by the Lena strain [14]. The PRRSV SD 16 and XJA1 strains used in this research are HP-PRRSV-2 strains, and the tissues were collected at 21 dpi. Taken together, these results indicate that PRRSV infection promotes LAG3 expression in vivo. We also examined the effects of PRRSV infection on the expression of two LAG3 ligands, MHC II and FGL1. The mRNA and protein expression of SLA-DRα in the four tissues and FGL1 in liver tissue also increased (Figure 4D–G). Considering the results of elevated LAG3 expression in the four tissues, we propose that PRRSV infection activates the FGL1/MHC II-LAG3 pathway. Therefore, the regulation of the LAG3-signaling pathway may affect PRRSV replication. To confirm this hypothesis, we first induced porcine monocytes into MoDCs (Figure 5) and then co-cultured PRRSV-infected MoDCs with activated porcine lymphocytes that overexpressed LAG3 or LAG3 expression interference with by siRNA, or with activated lymphocytes treated with mAb 1C2. By detecting the PRRSV N gene copy number and N protein expression in co-cultured cells, and virus titer in the supernatant, we determined that lymphocyte overexpression of LAG3 increased PRRSV replication in the co-culture system while interfering with LAG3 expression or blocking LAG3 signaling with mAb 1C2 inhibited PRRSV replication (Figure 6, Figure 7 and Figure 8). These results further supported the hypothesis that PRRSV infection activates the LAG3-inhibitory signaling pathway.

In this study, we observed an interesting phenomenon. When *pLAG3* expression was detected by Western blotting, although the same lysis buffer and protease inhibitors were used, the size of *pLAG3* expressed by pcDNA3.1/V5-HisB-flpLAG3-transfected HEK-293T cells was approximately 67 kDa (Figure 2B), whereas the size of *pLAG3* detected in mock and PRRSV-infected pig tissues was approximately 53 kDa (Figure 4C). This was true even when lysis buffer and protease inhibitors were replaced. Therefore, we believe that this was not due to lysis. Kim et al. prepared antisera by immunizing mice with recombinant *pLAG3* containing D1 and D2 structural domains expressed in CHO cells, and when PHA-stimulated porcine splenocytes were examined by Western blotting, the molecular mass of LAG3 detected was about 67 kDa [21]. Li et al. reported that mature LAG3 in mice is cleaved by metalloproteinases and broken into a 54 ku soluble p54 extracellular fraction and a 16 ku p16 transmembrane-cytoplasmic fraction. LAG3 must be cleaved from the cell surface to allow for normal T-cell activation, as non-cleavable LAG3 mutants prevent proliferation and cytokine production [19]. The extracellular region of *pLAG3* is composed of 435 amino acids (GenBank ID: NP_001098776.1), whereas that of mice is composed of 442 amino acids (GenBank ID: NP_032505.1). The size of *pLAG3* detected in pig tissues was approximately 53 kDa. This may indicate that *pLAG3* is cleaved by metalloproteinases in vivo to promote a normal T-cell immune response in pigs. However, this requires further investigation.

## 5. Conclusions

LAG3 is primarily responsible for transducing inhibitory signals. PRRSV infection increased the expression of LAG3 in vivo, and according to a previous study on serum FGL1 levels, we confirmed that PRRSV infection activates the FGL1/SLA II-LAG3 pathway. In future studies, we will explore whether PRRSV infection inhibits T cell activation through the FGL1–LAG3 pathway, which leads to immunosuppression and reduced viral clearance, thereby resulting in persistent infection in pigs.

## 6. Patents

Yang Mu, Xu Zheng, Rihua Cong. Preparation method of porcine LAG3 monoclonal antibody and its application. 4 November 2022, China, 202211375544.7.

## Figures and Tables

**Figure 1 vetsci-11-00483-f001:**
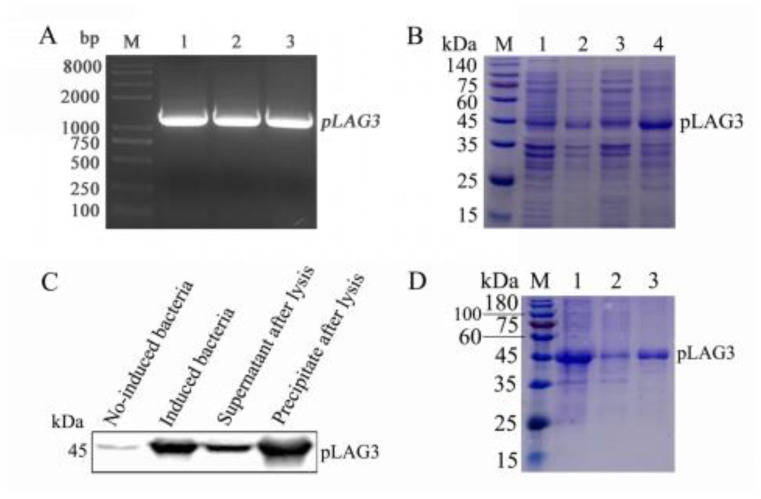
Amplification, expression, identification, and purification of recombinant *pLAG3*. (**A**) Amplified *pLAG3* (109–1305 nt) gene. M, DL8000 DNA marker; 1–3, *pLAG3* gene. (**B**) Identify *pLAG3* expression by SDS-PAGE and (**C**) Western blotting analysis. M, PageRuler Prestained Protein Ladder; 1, no-induced BL21-pET28a-pLAG3; 2, induced BL21-pET28a-pLAG3; 3, supernatant after lysis; 4, precipitate after lysis. (**D**) Purification of *pLAG3*. M, PageRuler Prestained Protein Ladder; 1, sample loaded; 2, effluent; 3, purified *pLAG3*.

**Figure 2 vetsci-11-00483-f002:**
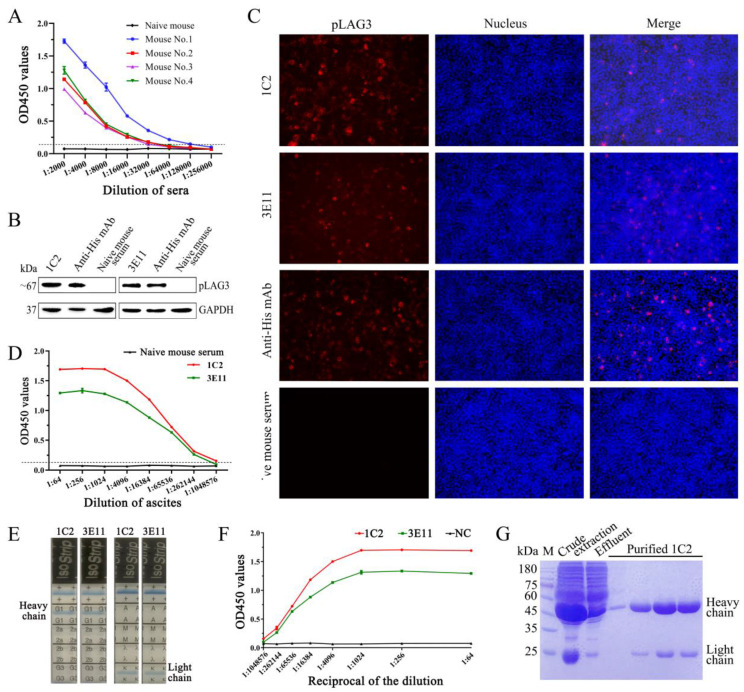
Characteristics of *pLAG3* mAbs. (**A**) Titers of mouse sera antibody against *pLAG3* detected by iELISA. Using anti-His mAb as a positive control, naive mouse serum as a negative control, mAb 1C2 or 3E11 as primary antibody, the reaction of mAbs 1C2 and 3E11 with *pLAG3* expressed by HEK-293T cells was determined by Western blotting (**B**) and indirect immunofluorescent assay (IFA) (**C**). (**D**) Titers of mAbs 1C2 and 3E11 in ascites detected by iELISA. (**E**) Isotype of mAbs 1C2 and 3E11 determined with an IsoStrip^TM^ Mouse Monoclonal Antibody Isotyping Kit (Cat# 11493027001, Roche). (**F**) Saturation curves of mAbs 1C2 and 3E11 with *pLAG3*. Per-well plates were coated with 200 ng purified recombinant *pLAG3*; mAbs 1C2 and 3E11 diluted differently were used to perform iELISA, and mAb2-5G2 was used as a negative control. (**G**) Purified mAb 1C2 identified by SDS-PAGE.

**Figure 3 vetsci-11-00483-f003:**
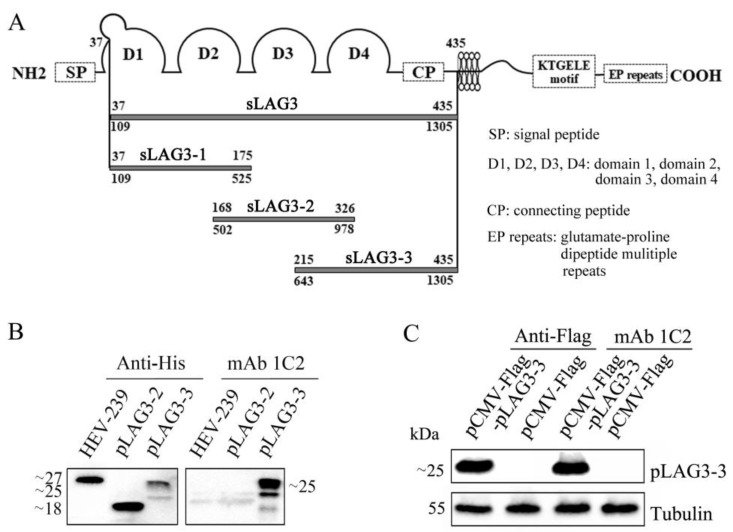
Identification of the antigen region recognized by mAb 1C2. (**A**) Schematic diagram of *pLAG3* fragments. (**B**) Procaryotic expression of truncated *pLAG3* and the reaction of mAb 1C2 with the truncated *pLAG3* identified by Western blotting. Porcine HEV 239 protein is a control protein expressed with the same vector and *E. coli* cells. (**C**) pLAG3-3 expressed by HEK-293T cells and the reaction of mAb 1C2 with pLAG3-3 identified by Western blotting.

**Figure 4 vetsci-11-00483-f004:**
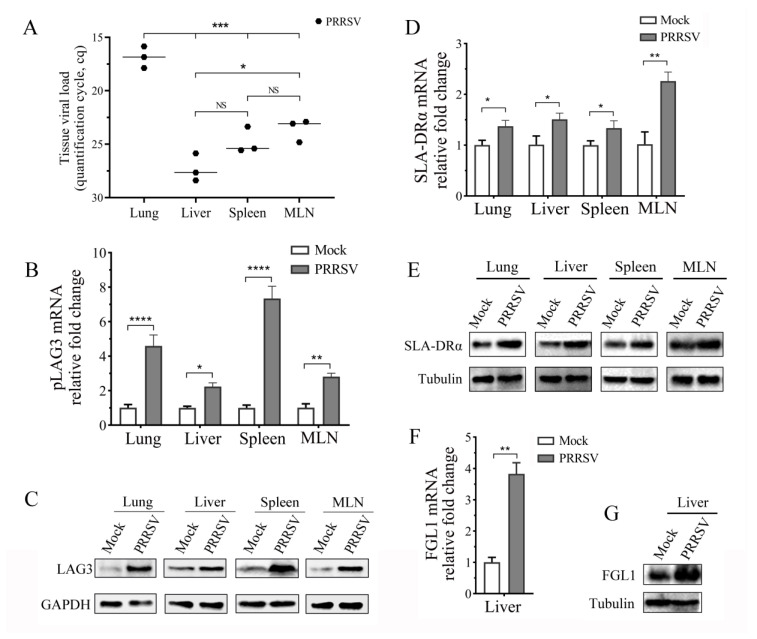
PRRSV infection promotes LAG3, SLA-DRα, and FGL1 expression in vivo. (**A**) PRRSV load, (**B**) *pLAG3*, (**D**) SLA-DRα, and (**F**) FGL1 gene expression were detected by RT-qPCR. The total RNA of lung, liver, spleen, and MLN samples from mock and PRRSV-infected pigs were extracted. Serial 10-fold dilutions of a recombinant plasmid containing 144 bp PRRSV ORF7 gene were introduced in each plate to detect inter-run variations, and the PRRSV load was shown as quantification cycle (Cq) value. Gene expressions were calculated with the 2^−ΔΔCt^ method, and GAPDH was used as an internal control. The results are shown as mean ± SD from three independent experiments. NS means no significant difference, * *p* < 0.05, ** *p* < 0.01, *** *p* < 0.001, **** *p* < 0.0001. (**C**) *pLAG3*, (**E**) SLA-DRα, and (**G**) FGL1 protein expressions in different tissues were detected by Western blotting, using homemade mAbs 1C2, 2E11D9, or 4G7 as the primary antibody.

**Figure 5 vetsci-11-00483-f005:**
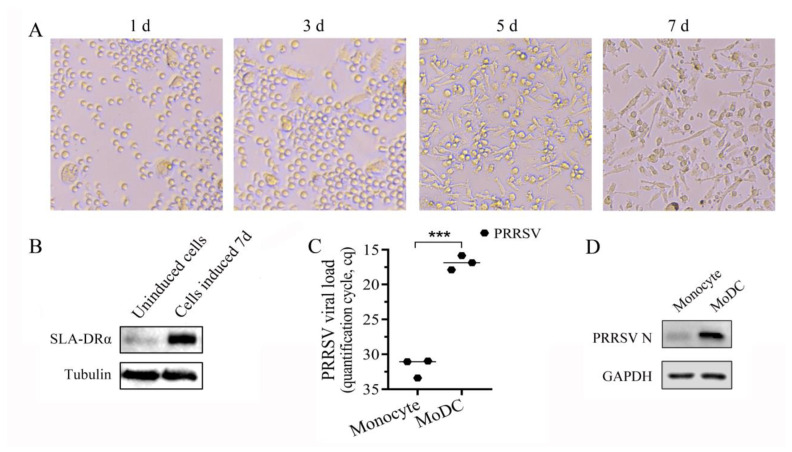
Induction of MoDCs in vitro. (**A**) Changes in cell morphology during MoDCs induction under a light microscope. (**B**) SLA-DRα expression was detected by Western blotting using homemade mAb 2E11D9. (**C**) PRRSV viral load was detected by RT-qPCR with 1 MOI PRRSV SD16 infection at 24 h. Serial 10-fold dilutions of a recombinant plasmid containing 144 bp PRRSV ORF7 gene were introduced in each plate to detect inter-run variations, and the PRRSV load was shown as quantification cycle (Cq) value. Gene expressions were calculated with the 2^−ΔΔCt^ method, and GAPDH was used as an internal control. The results are shown as mean ± SD from three independent experiments. *** *p* < 0.001. (**D**) PRRSV N protein expression was detected by Western blotting, using homemade mAbs 6D10 as the primary antibody at 24 hpi.

**Figure 6 vetsci-11-00483-f006:**
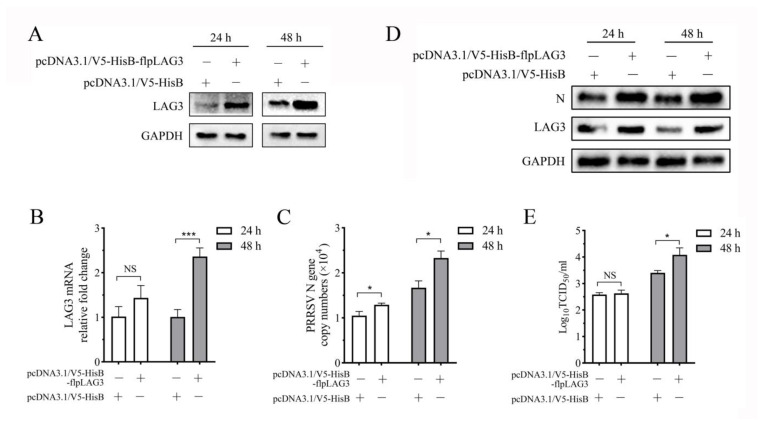
Lymphocyte overexpression LAG3 promotes PRRSV replication in the co-culture system. (**A**) The expression levels of LAG3 expression at 24 and 48 hpt were detected by Western blotting. LAG3 mRNA expression (**B**) and PRRSV N gene copy number (**C**) in the co-culture system were quantified by RT-qPCR. The results are shown as mean ± SD from three independent experiments. NS means no significant difference, * *p* < 0.05, *** *p* < 0.001. (**D**) LAG3 and PRRSV N protein expression in the co-culture system were detected by Western blotting. (**E**) The detection result of virus titer in the co-culture supernatant.

**Figure 7 vetsci-11-00483-f007:**
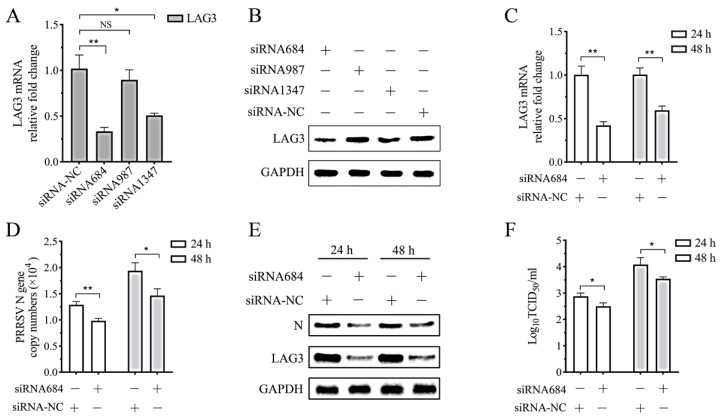
PRRSV replication was inhibited when interference LAG3 expression in lymphocytes with siRNA. The interference effect of specific siRNA was detected by (**A**) RT-qPCR and (**B**) Western blotting. The results are shown as mean ± SD from three independent experiments. NS means no significant difference, * *p* < 0.05, ** *p* < 0.01. (**C**) LAG3 gene expression and (**D**) PRRSV N gene copy number in the co-culture system were analyzed by RT-qPCR. (**E**) LAG3 and PRRSV N protein expression in the co-culture system were analyzed by Western blotting. (**F**) The detection result of virus titer in the co-culture supernatant.

**Figure 8 vetsci-11-00483-f008:**
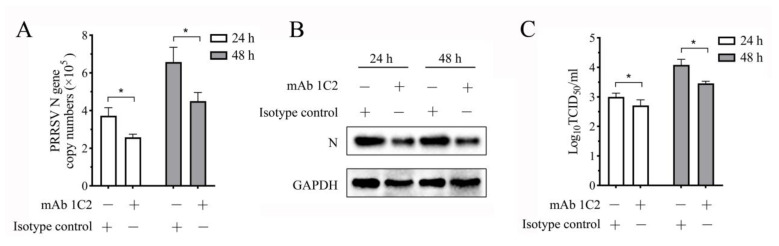
mAb 1C2 blocking inhibited PRRSV replication in the co-culture system. (**A**) RT-qPCR detection results of PRRSV N gene copy number in the co-culture system. The results are shown as mean ± SD from three independent experiments. * *p* < 0.05. (**B**) Western-blotting results of PRRSV N protein expression in the co-culture system. (**C**) The detection result of virus titer in the co-culture supernatant.

**Table 1 vetsci-11-00483-t001:** Primers’ information.

Primers’ Name	Sequence (5′-3′)	Target Fragment/bp
pLAG3-109-peF	CG***GGATCC***ATGGGGGCTCCTGC (*Bam*H I) ^a^	1200
pLAG3-1305-peR	CCC***AAGCTT***GAGGTGGCCTGTTTTCTGG (*Hin*d III)	
pLAG3-eeF	CCC***AAGCTT***ATGAGGGAGGCTCACTTCC (*Hin*d III)	1521
pLAG3-eeR	CG***GGATCC***GAGCTGCTCTGGCTG (*Bam*H I)	
pLAG3-109-peF	CG***GGATCC***ATGGGGGCTCCTGC (*Bam*H I)	417 (pLAG3-1)
pLAG3-525-peR	CCC***AAGCTT***GTTCAAAATGACCCAATGC (*Hin*d III)	
pLAG3-502-peF	CG***GGATCC***ATGACCTCGCATTGGGTCATT (*Bam*H I)	477 (pLAG3-2)
pLAG3-978-peR	CCC***AAGCTT***CTGCTGCCCCTGAAGATG (*Hin*d III)	
pLAG3-643-peF	CG***GGATCC***ATGCAAATCAGCCCCTTGGAC (*Bam*H I)	663 (pLAG3-3)
pLAG3-1305-peR	CCC***AAGCTT***GAGGTGGCCTGTTTTCTGG (*Hin*d III)	
pLAG3-643-eeF	CCC***AAGCTT***ATGCAAATCAGCCCCTTGGAC (*Hin*d III)	663 (pLAG3-3)
pLAG3-1305-eeR	CG***GGATCC***GAGGTGGCCTGTTTTCTG (*Bam*H I)	
pLAG3-qF	ACCTGGCAACCCGAAGAAA	217
pLAG3-qR	GCGAGACAGCTCCGTGAAGTA	
PRRSV-ORF7-qF	TAAGATCATCGCCCAACAAAAC	145
PRRSV-ORF7-qR	ACACAATTGCCGCTCACTAGG	
pGAPDH-qF	GGTGAAGGTCGGAGTGAACG	153
pGAPDH-qR	CCGTGGGTGGAATCATACTG	

Note: ^a^ The bold italics are *Bam*H I or *Hin*d III restriction enzyme sequences.

**Table 2 vetsci-11-00483-t002:** The sequences of siRNA.

siRNA Name		Sequence (5′-3′)	Target
siRNA684	Sense	CCUACAGAGAUGGCUUCAATT	*pLAG3*
	Anti-sense	UUGAAGCCAUCUCUGUAGGTT	
siRNA987	Sense	CUGUCACUUUGGCAGUCAUTT	*pLAG3*
	Anti-sense	AUGACUGCCAAAGUGACAGTT	
siRNA1347	Sense	CUGGAGCUAUUAGCUUUCATT	*pLAG3*
	Anti-sense	UGAAAGCUAAUAGCUCCAGTT	
NC	Sense	UUCUCCGAACGUGUCACGUTT	None
	Anti-sense	ACGUGACACGUUCGGAGAATT	

**Table 3 vetsci-11-00483-t003:** Results of ELISA additivity test.

Antibody	OD450 Values	AI	Possible Match
1C2 (1:1000)	1.148 ± 0.040	-	-
3E11 (1:1000)	1.202 ± 0.059	-	-
1C2 (1:1000) + 3E11 (1:1000)	1.216 ± 0.007	3.50%	<50%

## Data Availability

The datasets presented in this study can be found in online repositories. The names of the repositories and accession numbers are as follows: https://www.ncbi.nlm.nih.gov/nuccore/MK813967 (accessed on 9 January 2020); https://www.ncbi.nlm.nih.gov/protein/NP_001098776.1 (accessed on 30 June 2021); https://www.ncbi.nlm.nih.gov/protein/NP_032505.1?report=graph (accessed on 30 June 2021).

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
