# Peer review of "Monoclonal Antibody against Porcine LAG3 Inhibits Porcine Reproductive and Respiratory Syndrome Virus Infection"

_vetsci, 2024, doi:10.3390/vetsci11100483_

Round 1

Reviewer 1 Report

Comments and Suggestions for Authors

The manuscript describes work which has been executed to a high standard and is both novel and interesting. The introduction is comprehensive and covers the background work in the field in a balanced way. The methods used by the authors are described in sufficient detail for the work to be repeated by others if necessary. The results are presented in depth and the statistical analysis of the data appears to be highly rigorous. The discussion and conclusions summarise the findings and place them in a broader context as well as providing suggestions for future work. Overall the standard of presentation is high so I have no hesitation in recommending publication of the manuscript in its present form. 

One minor point is that some of the figure legends are quite brief e.g. Fig. 2 (C) is simply given as "IFA" and to this non-specialist referee the interpretation of these images defeated me. I could not find an explanation in the text either. I think it would be helpful to the reader have more explanation of these items. 

Author Response

Comments 1:

The manuscript describes work which has been executed to a high standard and is both novel and interesting. The introduction is comprehensive and covers the background work in the field in a balanced way. The methods used by the authors are described in sufficient detail for the work to be repeated by others if necessary. The results are presented in depth and the statistical analysis of the data appears to be highly rigorous. The discussion and conclusions summarise the findings and place them in a broader context as well as providing suggestions for future work. Overall the standard of presentation is high so I have no hesitation in recommending publication of the manuscript in its present form. 

Response: Thank you very much for your approval of our article.

Comment 2:

One minor point is that some of the figure legends are quite brief e.g. Fig. 2 (C) is simply given as "IFA" and to this non-specialist referee the interpretation of these images defeated me. I could not find an explanation in the text either. I think it would be helpful to the reader have more explanation of these items. 

Response: In agreement with the reviewer. We are sorry about that. We revised the description as “Using anti-His mAb as a positive control, naive mouse serum as a negative control, mAb 1C2 or 3E11 as primary antibody, the reaction of mAbs 1C2 and 3E11 with pLAG3 expressed by HEK-293T cells was determined by western blotting (B) and indirect immunofluoreseent assay (IFA) (C).” in lines 344-347 on page 9 of the revised manuscript.

Reviewer 2 Report

Comments and Suggestions for Authors

In the manuscript "Monoclonal Antibody Against Porcine LAG3 Inhibits Porcine 2 Reproductive and Respiratory Syndrome Virus Infection", Wang et al, developed and screened mAbs against porcine 13 LAG3 (pLAG3). The authors identified a candidate mAb (1C2) that showed good reactivity with pLAG3. The authors reported an increase of LAG3  in the tissues of Porcine reproductive and respiratory syndrome (PRRS)-infected pigs.  The main finding of this study is  the efficient inhibition of PRRS proliferation by mAb 1C2, highlighting the importance of  the LAG3 signaling pathway in PRRSV pathogenesis.

Please see below my comments and suggestions that could improve the overall quality of the manuscript.

Abstract: 

The abstract should contain enough information to provide a meaningful insight of the research. For that please consider my suggestions:

- Indicate in which conditions exactly the interaction between fibrinogen-like protein 1 and LAG3 is regarded as a new immune escape mechanism.

- Define the nature of PRRS (viral infection) and its clinical presentations.

- lines 27-28: in "showed good reactivity with pLAG3 27 and PHA-activated porcine lymphocytes", I guess it should be "on PHA-activated lymphocytes".

- state the source of lymphocytes - peripheral blood?

-lines 30-31: When stating that "results revealed 30 that PRRSV infection caused a marked increase in LAG3 expression", the increase of LAG3 was noted compared to control group? 

Introduction

Overall, the cohesion of the Introduction could be improved. There is a lot of information about LAG3 that should be organized in more concise way.  Only a small part focuses on PRRSV so the impact of the study is not emphasized. Authors mentioned the study of Ruedas-Torres about PRRSV-1 and it would be more clear if the PRRSV was introduced before, including the type of virus and pathology as well as consequences. 

- line 44: add a reference for LAG3 expression

- line 55: please rephrase this sentence for clarity: "LAG3 is also considered a marker of immunosuppressive regulatory T cell subpopulation activation". I think "immunosuppressive" could be omitted here.

- line 59: please clarify which type of T cells you are referring to. 

- line 62: reference is missing after "LAG3 is the third inhibitory receptor to be targeted in the clinic and is the most promising immune checkpoint after CTLA-4 and PD-1"

- line 66: the abbreviation should be introduced in the main text for PRRSV-1.

- I suggest moving this part: "MHC II, liver sinusoidal endothelial cell lectin, galactose lectin 3, and α-synuclein are the ligands identified for LAG3 [11]. Fibrinogen-like protein 1 (FGL1), a member of the fibrinogen family and a specific hepatocyte mitogen, has been reported to be the main  inhibitory ligand of LAG3 in the FGL1-LAG3 pathway, presenting a synergistic effect with  the PD-1/programmed death ligand 1 pathway, and is considered a promising immunotherapeutic target" to the line 47 before mouse knock-out model, for better coherence. 

- line 102: This sentence need clarification - PRRSV can cause immunosuppressive diseases.

Materials and Methods

- Please explain how exactly pigs were infected (volume of intramuscular injection, the description of intranasal administration). How many pigs were in the mock group and how many were in infected group. What was the sex of the pigs used in experiments?  

- What was the age and sex of mice used?

Results

- Figure 2A: consider using the term naive instead non-immunized mouse. Also, please color-code the individual mice as from the graph it is not clear which label belongs to which animal. 

- lines 385 -386: was the increase of viral load in the lungs significantly higher than in other organs? Please provide a p value. 

- Figure 5C: Please provide the p value as it was indicated in the text that the presented change is significant.

Comments on the Quality of English Language

Moderate editing of English language required

Author Response

Thank you very much for taking the time to review this manuscript. Please see the attachment to find the detailed responses.

Reviewer 3 Report

Comments and Suggestions for Authors

The draft of “Monoclonal Antibody Against Porcine LAG3 Inhibits Porcine Reproductive and Respiratory Syndrome Virus Infection” generate a monoclonal antibody against LAG3 that is an inhibitory receptor of T cells activation and proliferation. further results showed that PRRSV infection enhanced the LAG3 expression. Knocking down of LAG3 expression on PHA-activated lymphocytes promoted PRRSV replication,  and overexpression of LAG3 or blocking of the LAG3 signal with mAb inhibited PRRSV replication. All the data indicated that LAG3 signaling pathway plays an important role in the PRRSV infection in pigs. Overall the finding of this study shed lights on the role of porcine LAG3 in PRRSV infection.

Comments

1. The main target cell of PRRSV is macrophage cells, while the LAG3 is mainly expressed on the surface of T cell, how does the PRRSV regulate the LAG3 expression should be discussed.

2.  In the Co-Culture System, lymphocytes and DC cells were Co-Cultured for PRRSV infection study, why are those DC cells included in this system,but not macrophage cells.

3. In the result part of PRRSV Infection Promotes LAG3, SLA-DRα, and FGL1 Expression In Vivo ,the T cells activation and proliferation situation should be evaluated in the PRRSV infected pigs.

Author Response

Comment 1

The draft of “Monoclonal Antibody Against Porcine LAG3 Inhibits Porcine Reproductive and Respiratory Syndrome Virus Infection” generate a monoclonal antibody against LAG3 that is an inhibitory receptor of T cells activation and proliferation. further results showed that PRRSV infection enhanced the LAG3 expression. Knocking down of LAG3 expression on PHA-activated lymphocytes promoted PRRSV replication,  and overexpression of LAG3 or blocking of the LAG3 signal with mAb inhibited PRRSV replication. All the data indicated that LAG3 signaling pathway plays an important role in the PRRSV infection in pigs. Overall the finding of this study shed lights on the role of porcine LAG3 in PRRSV infection.

Response: Thank you very much for your approval of our article.

Comment 2

1. The main target cell of PRRSV is macrophagecells, while the LAG3 is mainly expressed on the surface of T cell, how does the PRRSV regulate the LAG3 expression should be discussed.

Response: In agreement with the reviewer. To explore this problem, we establish a co-culture system consisting of porcine lymphocytes and MoDCs. In this co-culture system, PRRSV infects MoDCs. While MoDcs, as a kind of antigen presenting cell, can transmit signals to T cells to promote T cell activation. Activated CD + T cells can kill MoDCs infected by viruses. If the expression of LAG3 in T cells is affected by PRRSV infection, the viability of T cells will change, which affects the function of T cells. The titer of PRRSV in the co-culture system will change. Therefore, the LAG3 expression and PPRSV changes in the co-cultures can reflect this problem. Through transcriptional sequencing of co-culture system cells, we have identified some key genes that play a role, and we are also exploring the signaling pathway of PRRSV infection regulation of LAG expression now.

Comment 3

2. In the Co-Culture System, lymphocytes and DC cells were Co-Cultured for PRRSV infection study, why are those DC cells included in this system, but not macrophagecells.

Response: We isolated mononuclear cells from porcine peripheral blood, leaving monocytes attached to the wall and discarding the lymphocytes not attached to the wall. Then the monocytes were induced to differentiate into MoDCs by rp IL-4 and rp GM-CST.

Comment 4

3. In the result part of “PRRSV Infection Promotes LAG3, SLA-DRα, and FGL1 Expression In Vivo” ,the T cells activation and proliferation situation should be evaluated in the PRRSV infected pigs.

Response: In agreement with the reviewer. We have taken this into account. In the subsequent animal experiments of LAG3 antibodies, we design a test to determine the activation and proliferation of T cells and the abundance of LAG3+ cells in T cell subsets by flow cytometry and cytokine level detection.